

# Frequent new particle formation at remote sites in the temperate/boreal forest of North America

Meinrat O. Andreae[1,2,3], Tracey W. Andreae[1], and Florian Ditas[1,4]

[1]Max Planck Institute for Chemistry, Mainz, Germany
[2]Scripps Institution of Oceanography, La Jolla, California, USA
[3]Department of Geology and Geophysics, King Saud University, Riyadh, Saudi Arabia
[4]Hessian Agency for Nature Conservation, Environment and Geology, Wiesbaden, Germany

*Correspondence to*: Meinrat O. Andreae (m.andreae@mpic.de)

**Abstract.** The frequency and intensity of new particle formation (NPF) over remote forest regions in the temperate and boreal zones, and thus the importance of NPF for the aerosol budget and life cycle in the pristine atmosphere, remains controversial. Whereas NPF has been shown to occur relatively frequently at several sites in Scandinavia, it was found to be nearly absent at a mid-continental site in Siberia. To explore this issue further, we made measurements of aerosol size distributions between 10 and 420 nm diameter at two remote sites in the transition region between temperate and boreal forest in British Columbia, Canada. The measurements covered 23 days during the month of June 2019, at the time when NPF typically reaches its seasonal maximum in remote mid-latitude regions. These are the first such measurements in a near-pristine region on the North American continent. Although the sites were only 150 km apart, there were dramatic differences in NPF frequency and intensity between them. At the Eagle Lake site, NPF occurred daily and nucleation mode particle concentrations reached above 5000 cm$^{-3}$. In contrast, at the Nazko River site, there were only 6 NPF events in 11 days and nucleation mode particle concentrations reached only about 800 cm$^{-3}$. The reasons for this difference could not be conclusively resolved with the available data; they may include airmass origins, pre-existing aerosols, and the density and type of forest cover in the surrounding regions. Our results suggest that measurement campaigns in the remote forest regions of North America to investigate the role of NPF with a more comprehensive set of instrumentation are essential for a deeper scientific understanding of this important process.

## 1 Introduction

Uncertainty regarding the magnitude of aerosol direct and indirect radiative effects is the largest contributor to the persistent uncertainty of net radiative forcing, which drives global and regional climate change (Boucher et al., 2013; Seinfeld et al., 2016; Bellouin et al., 2020; Naik et al., 2021). Since this forcing is the difference between present-day and pre-industrial radiative effects, both need to be known accurately to assess present-day forcing and prognosticate future forcings. Because of the strong non-linearity of the aerosol effects on cloud microphysics and precipitation, this applies in particular to the aerosol indirect effects, also referred to as aerosol-cloud interactions (ACI) (Twomey et al., 1984; Rosenfeld et al., 2008; Carslaw et al., 2013; Carslaw et al., 2017). Adding the same amount of pollution aerosol to a pre-industrial atmosphere with very low initial particle concentration would cause a much greater radiative forcing than adding the same amount to an atmosphere with a higher pre-industrial background (Carslaw et al., 2013; Gordon et al., 2016; Hamilton et al., 2018).

Because the ACI are driven by the microphysical perturbations of cloud properties by aerosols, they are a function of the number concentration of cloud condensation nuclei (CCN), i.e., the subset of aerosol particles that can nucleate cloud droplets. It is thus essential to understand the sources and lifecycles of this particle class in both polluted and pristine

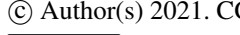



atmospheres. While some CCN are the result of physical processes that directly release particles into the atmosphere, e.g., seaspray and mineral dust, a large and possibly dominant fraction of CCN is the result of secondary production of particles from the condensation of trace vapors. Models suggest that around 40-70% of global CCN originate from nucleation of gaseous compounds and subsequent growth of the nucleated embryos into aerosol particles, a process called new particle formation (NPF) (Merikanto et al., 2009; Wang and Penner, 2009; Yu and Luo, 2009; Zhang et al., 2012; Dunne et al.,

2016; Gordon et al., 2017; Kerminen et al., 2018). The importance of NPF for the global CCN budget may be higher in pristine than anthropogenically perturbed atmospheres: Gordon et al. (2017) estimated a contribution by NPF of 67% of CCN for the preindustrial atmosphere in contrast to 54% at the present day.

Under present-day conditions, sulfuric acid ($H_2SO_4$) driven pathways are considered to dominate daytime nucleation, based on laboratory studies (Sipilä et al., 2010) and field measurements at numerous sites (Kerminen et al., 2018;

Nieminen et al., 2018), e.g., Hyytiälä, Finland (Ehn et al., 2010; Kulmala et al., 2013), the high Arctic (Giamarelou et al., 2016), and marine environments (Brean et al., 2021; Zheng et al., 2021). In these pathways, gaseous $H_2SO_4$ is required to form the initial clusters, which are stabilized synergistically by $NH_3$, amines, and organics, particularly highly oxidized molecules (HOMs) (Zhang et al., 2012; Kulmala et al., 2013; Schobesberger et al., 2013; Riccobono et al., 2014; Dal Maso et al., 2016; Kürten et al., 2018; Lehtipalo et al., 2018). In addition to these sulfuric acid driven pathways, models

and laboratory studies suggest that pure organic nucleation, possibly ion-induced, may be a significant source of new particles in present-day pristine and pre-industrial atmospheres (Jokinen et al., 2015; Gordon et al., 2016; Kirkby et al., 2016; Gordon et al., 2017; Zhu and Penner, 2019). This has been supported by observations at some remote upper tropospheric sites, e.g., in the Himalayas (Bianchi et al., 2021) and the Bolivian Andes (Rose et al., 2015).

In contrast to $H_2SO_4$ driven NPF, which most commonly happens in the first half of the day, pure organic nucleation can

also result from the nighttime oxidation of biogenic volatile organic compounds (BVOCs) by ozone or by autoxidation to form highly oxidized molecules (HOMs) with extremely low volatility. Nighttime nucleation has been observed to occur frequently in some environments with low condensation sinks (Vehkamäki et al., 2004; Lee et al., 2008; Suni et al., 2008). HOM dimer concentrations have their maxima during the night (Sulo et al., 2021), and HOMs from monoterpene oxidation have been observed to drive nighttime nucleation in springtime at Hyytiälä (Rose et al., 2018). Laboratory

studies and quantum chemical calculations demonstrate the important role of monoterpene oxidation products for nighttime nucleation (Ortega et al., 2012; Bianchi et al., 2019). Once new particles with diameters of a few nm have formed, they can grow by condensation of additional sulfates and organics. Particle growth at sites without high pollution levels is dominated by condensation of organics, mostly from the oxidation of biogenic VOCs (Riipinen et al., 2012; Ehn et al., 2014; Dal Maso et al., 2016; Tröstl et al., 2016; Bianchi et al., 2019).

Reviews of the worldwide distribution of NPF events have shown them to occur in all types of environments, from very remote to highly polluted (Kerminen et al., 2018; Nieminen et al., 2018; Bousiotis et al., 2021). Generally, NPF frequencies are highest at pollution-impacted sites, e.g., the Po Valley (Kontkanen et al., 2016), Mexico City (Dunn et al., 2004), and Beijing (Yan et al., 2021). However, NPF has also been shown to occur quite frequently at rural sites, for example Hyytiälä, Finland, where the highest frequency is observed in spring with 47% of days being NPF days (Dada et al., 2018;

Nieminen et al., 2018). The role that anthropogenic emissions play at these rural and some remote sites is still unclear. In a study at a remote site in northern Finland, Kyrö et al. (2014) showed that the frequency of NPF events declined with decreasing $SO_2$ emissions from a smelter in the region. Similarly, NPF was found to occur fairly frequently at two Siberian sites located in the boreal forest (Dal Maso et al., 2008), but the proximity of high-emitting urban and industrial regions (Tomsk and Irkutsk) makes the interpretation of these results problematic. This also applies to the few NPF studies in

North America at sites in the temperate forest zone. At Egbert, Ontario, Canada, nucleation occurred frequently, but $SO_2$ concentrations in the range of 1-3 ppb during event days are clear evidence of substantial anthropogenic input even under


relatively clean conditions for this site (Pierce et al., 2014). Similarly, $SO_2$ concentrations of 0.3 to 1 ppb indicate a significant anthropogenic influence at an isoprene-dominated site in the Ozark Mountains, Missouri, where frequent and intense NPF was observed (Yu et al., 2014). At Whistler, British Columbia, there was evidence for NPF on five days

during a period of atmospheric high pressure and elevated temperatures, but the presence of $SO_2$ in the range of 0.05 to 0.1 ppb suggests that anthropogenic emissions also played a role here (Pierce et al., 2012).

Unfortunately, there are very few NPF studies at truly remote continental sites, in part because it is quite difficult to find sites with near-pristine conditions on the continents (Andreae, 2007). The sites classified as remote in Nieminen et al. (2018) are all in regions where significant anthropogenic pollution can be expected, at least much of the time, e.g., Fi-

nokalia, Greece, and Mt. Waliguan, China. In contrast to these "remote" sites, where annual median NPF frequencies around 20% have been observed, NPF was found to be very rare during multi-year observations at very remote sites in Central Siberia (Wiedensohler et al., 2019; Uusitalo et al., 2021) and almost absent in the central Amazon Basin (Andreae et al., 2015; Rizzo et al., 2018; Wimmer et al., 2018; Franco et al., 2021). In the Amazon, NPF was, however, detected occasionally at sites affected by the pollution plume from Manaus, indicating the effect of even quite small amounts of

anthropogenic inputs on NPF (Rizzo et al., 2018; Wimmer et al., 2018).

The stark contrast between the observations of frequent NPF at the Scandinavian rural and remote sites, and its near-absence at our remote Siberian and Amazon sites prompted us to ask the question, whether NPF in the planetary boundary layer over vegetated land surfaces would only occur in the presence of at least a minor amount of recent anthropogenic $SO_2$ inputs, or if it could also happen in a truly pristine environment. In particular, we were interested to address this

question in an area comparable to the temperate-to-boreal forest environment that has been shown to be a prolific source of new particles in Northern Europe. Based on maps of aerosol optical depth (e.g., Huneeus et al., 2012), vegetation types, topography, and potential anthropogenic sources we identified the interior of British Columbia, Canada, as a suitable region for a pilot study. Measurements in this region seemed especially important in view of the fact that no studies on NPF had been conducted at any near-pristine site in North America.

Here, we present the results of measurements of aerosol size distributions in the range between 10 and 420 nm diameter at two remote sites in the transition region between temperate and boreal forest in British Columbia, Canada, collected over 23 days during the month of June 2019. From these measurements, we derived estimates of NPF frequency, particle growth rate, and condensation sink.

## 2 Methods

### 2.1 Measurement sites

Our measurements were conducted at two remote locations in the Fraser River basin of British Columbia, Canada (Fig. 1a). The basin lies between the Coast Mountains to the West and the Rocky Mountains to the East and has a generally gentle topography, dominant forest cover, and very low population density ($<0.5$ km$^{-2}$). The few larger towns are along the Fraser River and Highway 97, far downwind (70 to 160 km) of our sites during the prevailing westerly winds. Back-

trajectory analysis showed that the sampled airmasses did not contact this inhabited region.

### 2.1.1 Eagle Lake

The Eagle Lake (EL) site (51.90 ºN, 124.38 ºW, 1066 m a.s.l.) is on a small, isolated ranch surrounded by large tracts of predominantly evergreen forest (Fig. 1b). The instrumentation was deployed in a small cabin west of the ranch house. Access is by a rough road that dead-ends at the ranch. A small lake (Eagle Lake) lies northwest of the site. A few cabins

are located about 4 km away on the other side of the lake. Electric power was provided by a small hydroelectric generator.





The aerosol inlet was located about 2 m above ground level, and the sample air was brought into the cabin by ca. 2 m of 6.25 mm OD copper tubing. Tests comparing measurements with and without the inlet tubing at times with a pronounced nucleation mode showed no detectable particle loss. The air was sampled without the use of a dryer.

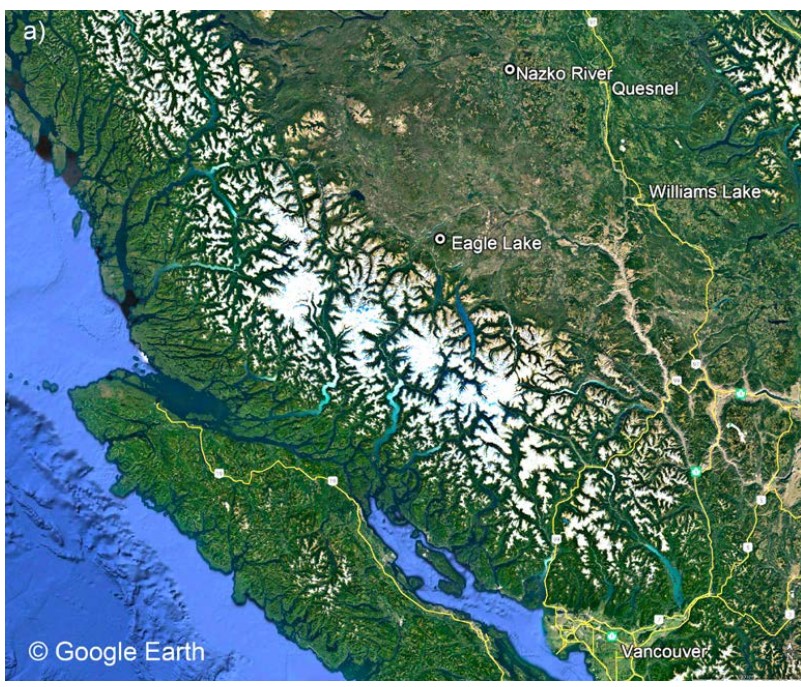


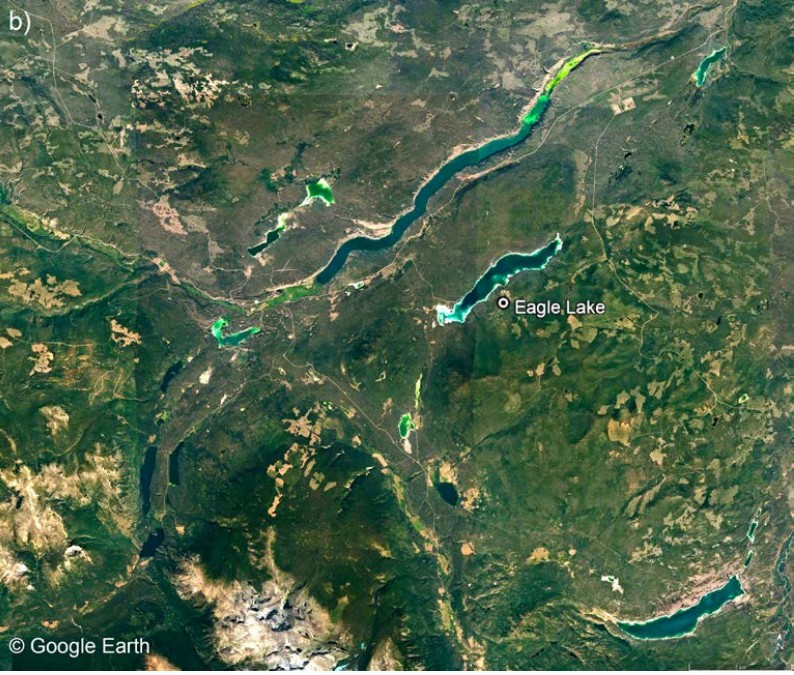





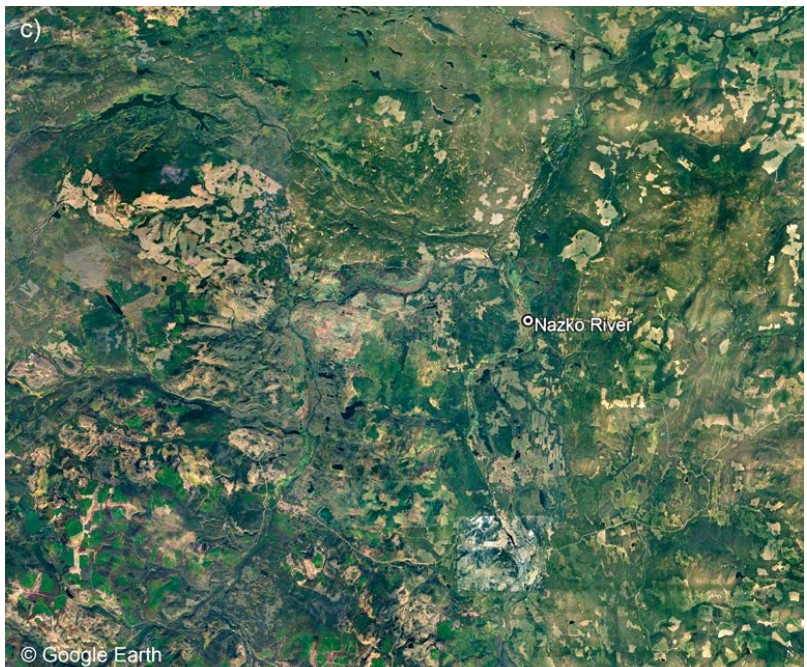

**Fig. 1: a) Overview map of the sampling area. b) detailed view of the area surrounding the Eagle Lake site. c) detailed view of the area surrounding the Nazko River site (Map base courtesy of Google Earth).**

### 2.1.2 Nazko River

The Nazko River (NR) site consists of a small isolated cabin (53.08 ºN, 123.56 ºW, 840 m a.s.l.) on the western bank of the Nazko River (Fig. 1c). The fetch region is characterized by a patchwork of evergreen forest and open areas with regrowth of small pines, abundant small and medium-sized aspen, and weedy vegetation. A small gravel road runs about 200 m west of the site, with a traffic volume of a few tens of vehicles per day. No response from the few passing cars could be detected in the data. There is no human habitation for at least 80 km in the upwind fetch of the site. Electricity was provided by line power. The instrument was located in a small shed upwind of the cabin, with an inlet layout similar to that at EL.

### 2.2 Instrumentation

The measurements of aerosol number size distribution were made with a Nanoscan 3910 SMPS Particle Sizer (TSI Inc., Shoreview, Minn., USA). The Nanoscan 3910 uses a unipolar corona charger, a radial differential mobility analyzer, and an isopropanol-based Condensation Particle Counter (CPC). It measures nanoparticle size distributions from 10 to 420 nm (13 channels) in one minute intervals. The response is linear between $10^2$ and $10^6$ cm$^{-3}$, and the sizing accuracy is better than 8%. Based on the manufacturer, particle diameters and concentrations agree to within 5% with measurements made using a TSI SPMS 3936 particle sizer, and the reproducibility for median particle diameter and total concentration are 1.1% and 2.7%, respectively (TSI Inc. Application Note NANOSCAN-002). An independent study showed similar agreement, with deviations of particle size $\leq$4% and concentration $\leq$10% (Vo et al., 2018). The instrument was calibrated by the manufacturer before the field deployment. For the calculation of aerosol mass concentrations, the density of 1.2 g cm$^{-3}$ specified by the manufacturer was used. Concentration data are reported relative to ambient temperature and pressure.



### 2.3 Meteorological information


Meteorological data were obtained from the NARR (North American Regional Reanalysis) database using the NOAA - READY (National Ocean and Atmospheric Administration - Real-time Environmental Applications and Display sYstem) tool (Rolph et al., 2017). The NARR data is available on a 3-hourly, 32-km grid. The NARR project is an extension of the NCEP Global Reanalysis, which is run over the North American Region. The NARR model uses the very high resolution NCEP Eta Model (32-km/45 layer) together with the Regional Data Assimilation System (RDAS), which assimilates precipitation along with other variables. Backward trajectories (BT) were computed using the NOAA Hybrid Single-Particle Lagrangian Integrated Trajectory model (HYSPLIT) with meteorological input data from the Global Data Assimilation System (GDAS, 1° resolution) (Stein et al., 2015).


### 2.4 Ancillary data


Concentrations of seasalt, dust, and black carbon (BC) aerosol for the study region were obtained from the Modern-Era Retrospective analysis for Research and Applications, Version 2 (MERRA-2) data base using the Giovanni online data system (https://giovanni.gsfc.nasa.gov).

## 3 Results and discussion

### 3.1 Meteorological background


The meteorological conditions at both sites during the study periods are summarized in Figs. 2a and 2b. Overall, conditions at both sites were mostly fair or partly cloudy, with abundant sunshine and occasional light showers. Overcast skies were encountered occasionally, most frequently in the morning or during the passage of showers. An extended period of overcast from 9 to 11 Jun at EL consisted mostly of high thin cloud. Daytime shortwave radiation levels showed mid-day maxima generally between 600 and 900 W m⁻².


The passage of a synoptic cycle at EL resulted in dominant low pressure at the beginning and end of the study period, with a high-pressure system in the middle of the period. This resulted in an overall dominance of fairly light (average 3.0 m s⁻¹, range 2.0-6.3 m s⁻¹) southwesterly to northwesterly winds, with a brief period of southeasterly winds when the pressure was falling on 11-12 Jun. Temperatures were initially low at EL, with highs around 10 ℃ and lows around 5 ℃, but increased to highs around 20 ℃ and lows around 10 ℃ after the passage of the high pressure system. Relative humidities were in the range of 24-84%, averaging 55%.


At NR, winds were predominantly northerly to northwesterly, with a brief period of southwesterly winds on 17 Jun. Wind speeds were slightly higher than at EL (average 3.9 m s⁻¹, range 3.2-8.5 m s⁻¹). Low temperatures were typically between 5 and 10 ℃ and highs between 13 and 21 ℃, and relative humidities were in the range of 29-93%, averaging 60%, slightly more humid than at EL.




**Fig. 2:** Meteograms for the study periods at the a) Eagle Lake and b) Nazko River sites. Wind direction and speed are indicated by barbs at the bottom of the plots.





**3.2 Airmass history**

The 48-h airmass backtrajectories initialized 100 m above surface level at 12 local time (LT = 19 UTC) are shown for the two sites in Figs. 3a and 3b (separate 72-h backtrajectories for each individual day are available in the Supplement). All trajectories for the Eagle Lake site had crossed the Pacific coast 22 to 48 hours before arriving at the site and then traveled across the densely forested Coast Range. This fetch area has an extremely low population density and is devoid of any

industrial activity. All but two of the trajectories travelled in the boundary layer for at least 48 hours before their arrival. The airmasses arriving on 12 and 13 Jun had not made surface contact with the ocean surface; instead, they arrived from the free troposphere over the Pacific, and descended rapidly after having crossed the Coast Range.

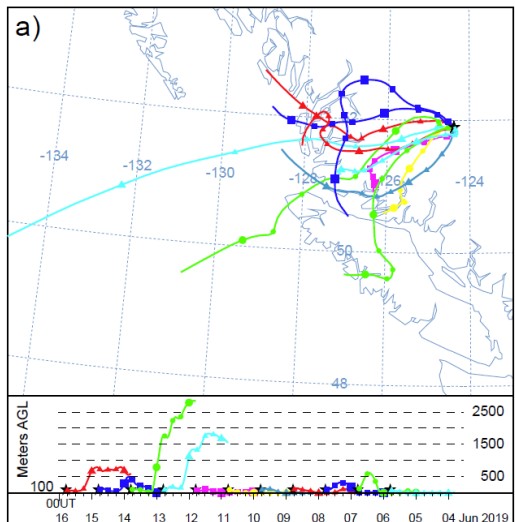

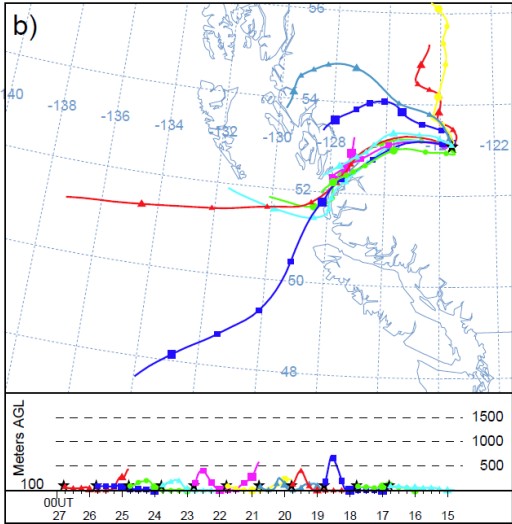

**Fig. 3: 48-hour backtrajectories initialized on each sampling day at 1900 UTC at 100 m above ground level at a) Eagle Lake and b) Nazko River.**





With two exceptions, the airmasses arriving at the Nazko River site also arrived from the Pacific coast, which they had crossed about 14 to 48 hours before arriving at NR. During and after crossing the Coast Range, they remained in the boundary layer for their entire travel. Similar to the fetch at EL, there is no industrial activity and very little human

population in this fetch area. Two airmasses arrived from the North and had not made contact with the ocean surface in the last 72 hours. These airmasses originated in an area that had been influenced by the smoke from fires that had been burning for several weeks in northern Alberta, and about 24 h before arriving at NR, they crossed the small municipality of Vanderhoof, which lies 108 km from the site and has a population of about 10,000 persons.

In summary, our analysis of the history of the sampled airmasses shows that they had no significant input of anthropogenic

emissions for at least 3 days before our measurements. An analysis of 10-day backtrajectories showed that even on this time scale, almost all airmasses had remained over the Pacific Ocean, with only a few trajectories crossing over remote regions of British Columbia or Alaska (Supplemental Figs. S1a and S1b).

### 3.3 Aerosol concentrations and size distributions

Figs. 4a and 4b show time series of the number concentrations of aerosol particles in the nucleation mode ($N_{nuc}$; 10 to 24

nm diameter) and across the entire size range covered by the Nanoscan SMPS ($N_{420}$; 10 to 420 nm), as well as the aerosol mass concentration in the 10 to 420 nm range ($M_{420}$) derived from the number spectra by assuming spherical particles with a density of 1.2 µg cm$^{-3}$. The corresponding size distributions are shown in Figs. 5a and b. The mass concentrations of dust, seasalt, and BC from the MERRA-2 models are provided in the supplement (Supplemental Figs. S2a and b).

The measurements at Eagle Lake indicate an extremely clean atmosphere: The time series plot of aerosol number size

distributions (Fig. 5a) is dominated by particles below 100 nm, often with a distinct nucleation mode below 20 nm and a separate Aitken mode between 20 and 80 nm. The accumulation mode tends to be absent or weak, except for a short period on 12 and 13 Jun, when a mode around 100 nm could be interpreted as a more pronounced accumulation mode. The mass concentrations, $M_{420}$, are extremely low (Fig. 4a), with an average of 0.73 (range 0.11 – 2.3) µg m$^{-3}$. The BC, dust, and seasalt concentrations from MERRA-2 are also very low, at 0.041±0.011, 2.9±1.5, and 0.41±0.37 µg m$^{-3}$, re-

spectively, for the measurement period (Supplemental Fig. S2a). In contrast, the particle number concentrations, $N_{420}$, are extremely high, with an average of 3150 (850 – 10300) cm$^{-3}$, of which a large fraction are in the nucleation mode below 24 nm (average 1100, range 55 – 5250 cm$^{-3}$) (Fig. 4a).





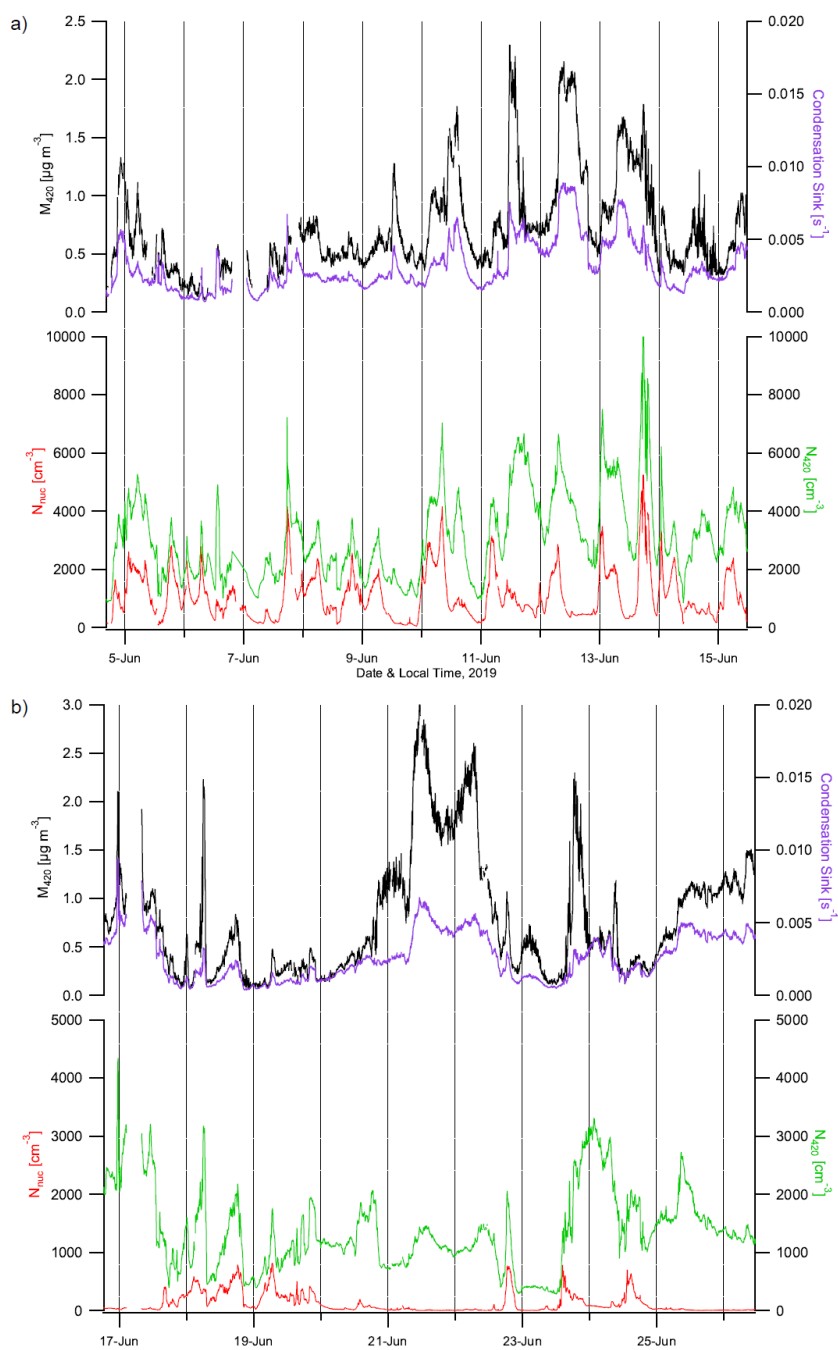

**Fig. 4: Mass and number concentrations of aerosol particles <420 nm ($M_{420}$ and $N_{420}$), number concentrations of nucleation mode particles <24 nm ($N_{nuc}$), and condensation sink at a) Eagle Lake and b) Nazko River.**

Like at EL, the BC, dust, and seasalt concentrations at the Nazko River site were very low, at 0.063±0.062, 1.3±0.8, and 0.34±0.35 µg m$^{-3}$, respectively (Supplemental Fig. S2b). The low BC concentrations indicate the near-absence of combustion-derived pollutants, with the exception of a period during 21-22 Jun, when the backtrajectories indicated transport from the north. During this episode, aerosol mass concentrations, BC, and the accumulation mode number concentrations





were slightly elevated (Fig. 4b and 5b), suggesting an influence of smoke from the large fires that were burning since mid-May in northern Alberta, about 600-800 km NE of the site. Excluding this period, BC concentrations averaged $0.036\pm0.017$ µg m$^{-3}$. The average $M_{420}$ concentrations at NR were similar to those at EL, with an average of $0.77\pm0.61$ µg m$^{-3}$, which is reduced to $0.57\pm0.40$ m$^{-3}$ (range $0.063 - 2.3$) when the smoke-affected period is excluded. The number

concentrations, $N_{420}$ and $N_{nuc}$, were $1350\pm670$ (range $280 - 4340$) and $123\pm165$ (range $3 - 820$) cm$^{-3}$, respectively, substantially lower than at EL.

a)

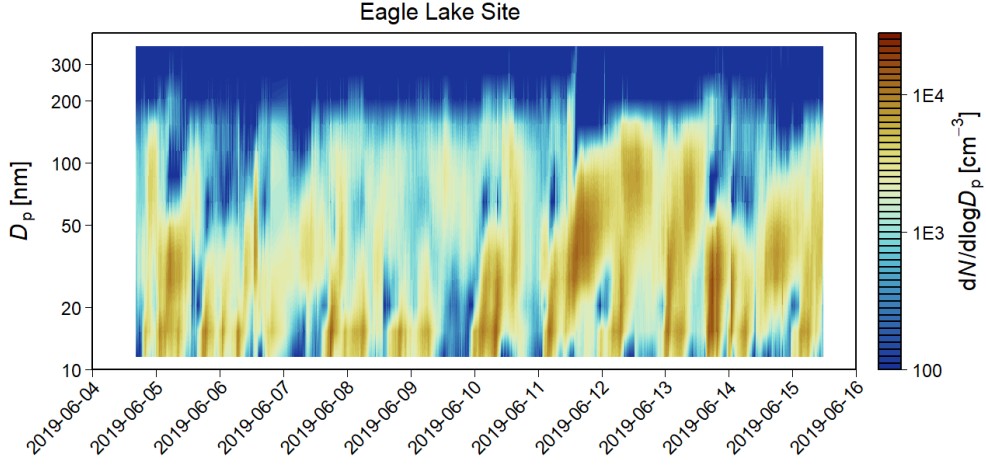

b)

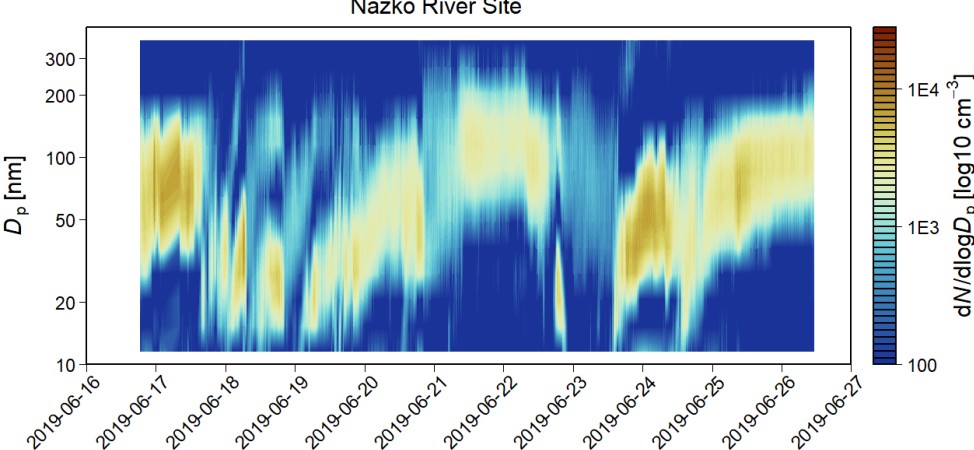


**Fig. 5: Time series of aerosol number size distributions at a) Eagle Lake and b) Nazko River.**



### 3.4 New particle formation events

### 3.4.1 NPF frequency

New particle formation events were identified using the procedure and criteria of Kulmala et al. (2012). The criteria for

an NPF event were: (1) a distinct increase of $N_{nuc}$ concentrations, (2) formation of a new nucleation mode persisting for more than two hours, and (3) growth of the nucleation mode over several hours. The NPF event identification was first performed by an automated routine (Franco et al., 2021) and then verified visually on the basis of the daily surface plots of the time series of the number size distributions and the time evolution of $N_{nuc}$ particles. A few events erroneously flagged as NPF by the automated algorithm were removed. The characteristics of the NPF events at our two sites are

summarized in Table 1.

**Table 1: New particle formation events at the Eagle Lake and Nazko River sites: Timing, condensation sink at the beginning of the event, formation rate at 10 nm, and growth rate.**

| Event Number | Event Start [Local Time] | Event End [Local Time] | Event Duration [min] | Condensation Sink [s$^{-1}$] | Formation Rate at 10 nm [cm$^{-3}$ h$^{-1}$] | Growth Rate [nm h$^{-1}$] |
|---|---|---|---|---|---|---|
| Eagle Lake | | | | | | |
| 1 | 04-Jun-19 18:40 | 04-Jun-19 21:00 | 140 | 0.0019 | 0.29 | 2.0 |
| 2 | 05-Jun-19 00:00 | 05-Jun-19 05:00 | 300 | 0.0046 | 0.35 | 1.8 |
| 3 | 05-Jun-19 15:40 | 05-Jun-19 20:00 | 260 | 0.0025 | 0.26 | 1.0 |
| 4 | 06-Jun-19 00:00 | 06-Jun-19 02:30 | 150 | 0.0010 | 0.33 | 2.4 |
| 5 | 06-Jun-19 06:30 | 06-Jun-19 10:00 | 210 | 0.0012 | 0.65 | 2.8 |
| 6 | 06-Jun-19 12:40 | 06-Jun-19 15:10 | 150 | 0.0014 | 0.40 | 4.6 |
| 7 | 07-Jun-19 15:30 | 07-Jun-19 19:35 | 245 | 0.0018 | 0.49 | 1.6 |
| 8 | 08-Jun-19 00:00 | 08-Jun-19 07:40 | 460 | 0.0027 | 0.07 | 0.2 |
| 9 | 08-Jun-19 15:30 | 08-Jun-19 21:00 | 330 | 0.0022 | 0.15 | 0.5 |
| 10 | 09-Jun-19 03:10 | 09-Jun-19 12:50 | 580 | 0.0019 | 0.09 | 1.0 |
| 11 | 09-Jun-19 22:30 | 10-Jun-19 04:00 | 330 | 0.0020 | 0.19 | 1.0 |
| 12 | 10-Jun-19 07:00 | 10-Jun-19 11:00 | 240 | 0.0033 | 0.47 | 3.6 |
| 13 | 11-Jun-19 02:00 | 11-Jun-19 08:30 | 390 | 0.0019 | 0.32 | 2.1 |
| 14 | 11-Jun-19 09:40 | 11-Jun-19 21:00 | 680 | 0.0026 | 0.22 | 2.5 |
| 15 | 12-Jun-19 03:50 | 12-Jun-19 12:00 | 490 | 0.0042 | 0.16 | 1.5 |
| 16 | 12-Jun-19 23:30 | 13-Jun-19 02:00 | 150 | 0.0029 | 0.43 | 3.5 |
| 17 | 13-Jun-19 05:00 | 13-Jun-19 08:50 | 230 | 0.0048 | 0.18 | 1.0 |
| 18 | 13-Jun-19 15:20 | 13-Jun-19 22:00 | 400 | 0.0042 | 0.59 | 2.1 |
| 19 | 13-Jun-19 23:30 | 14-Jun-19 02:00 | 150 | 0.0019 | 0.54 | 5.2 |
| 20 | 14-Jun-19 04:00 | 14-Jun-19 09:00 | 300 | 0.0016 | 0.16 | 1.9 |
| 21 | 15-Jun-19 02:00 | 15-Jun-19 07:00 | 300 | 0.0023 | 0.34 | 1.3 |
| | Average | | 309 | 0.0025 | 0.32 | 2.1 |
| | Standard Deviation | | 148 | 0.0011 | 0.16 | 1.3 |
| Nazko River | | | | | | |
| 22 | 17-Jun-19 15:00 | 17-Jun-19 20:00 | 300 | 0.0030 | 0.12 | 2.3 |
| 23 | 18-Jun-19 02:00 | 18-Jun-19 06:40 | 280 | 0.0006 | 0.07 | 4.0 |
| 24 | 18-Jun-19 08:00 | 18-Jun-19 14:30 | 390 | 0.0006 | 0.03 | 2.4 |
| 25 | 19-Jun-19 02:00 | 19-Jun-19 12:00 | 600 | 0.0006 | 0.05 | 1.3 |
| 26 | 23-Jun-19 14:00 | 23-Jun-19 17:00 | 180 | 0.0007 | 0.33 | 6.0 |



| 27 | 24-Jun-19 13:40 | 24-Jun-19 18:00 | 260 | 0.0014 | 0.11 | 3.3 |
|---|---|---|---|---|---|---|
| | Average | | 335 | 0.0012 | 0.12 | 3.2 |
| | Standard Deviation | | 146 | 0.0010 | 0.11 | 1.6 |

At Eagle Lake, we identified 21 distinct NPF events in a time span of only 12 days, with up to three events within a 24-h period (Table 1 and Fig. 5a). Using the classification of Kulmala et al. (2012), which focuses on event days rather than single NPF events, all 12 days were event days. We further classified the events into daytime and nighttime events using an adjusted event start time. Since the lower cutoff our instrument is 10 nm, it will detect an event some 3 hours later than an instrument with a cutoff at 3 nm would, given our average GR of 2-3 nm h$^{-1}$. For the purpose of classifying events as daytime or nighttime, we thus adjusted the start time by subtracting 3 hours from the observed start times listed in Table 1, which are based on detection with a lower cutoff at 10 nm. Events were classified as nighttime NPF, if the adjusted start time fell between 21:00 and 04:30 LT, i.e., the time when the SW radiation flux was typically below 20 W m$^{-2}$. Using this criterion, we found 11 nighttime events at EL, with nighttime nucleation occurring on 8 of 11 nights, and in some instances with two events in a single night.

The frequency of NPF was much lower at Nazko River (Fig. 5b), with only six events over 10 days, and five days classified as event days. Nighttime NPF also occurred less frequently, with only three events during the study period. During the smoke-affected period of 21-22 Jun, there was no evidence of NPF, and $N_{nuc}$ levels were very low, with an average of 16 cm$^{-3}$. Nucleation activity resumed immediately, however, around 18 LT on 22 Jun, when accumulation mode concentrations had dropped back down to background values.

Given the relatively short duration of our study compared to the long-term studies at some other sites, we can only make limited comparisons regarding NPF frequency at our BC sites. Our study took place during late spring, when NPF events are most frequent at other temperate/boreal sites, e.g., Hyytiälä (Dada et al., 2017) and Vavihill, Sweden (Kristensson et al., 2008). Similarly, at Pallas, a remote site at the northern edge of the boreal forest (68 ºN) in Finland, Asmi et al. (2011) found that NPF event frequency peaked in spring, while the highest growth rates occurred in summer. Taking both our sites together, we had 17 event days out of a total of 22 days of observations, for a frequency of 77%, which is significantly higher than the median frequencies of 20-50% reported for the spring/summer seasons at comparable sites by Nieminen et al. (2018). This is largely a consequence of the frequent nighttime NPF at our sites, because without the nighttime events there would be only 11 NPF days, or a frequency of 50%.

### 3.4.2 Diurnal behavior

New particle formation events occurred at our sites just as often during nighttime as during daytime (13 out of 27 events), in sharp contrast to what has been observed at most other sites. For example, NPF at Hyytiälä typically takes place between sunrise and noon (Dada et al., 2017), and at Egbert, Ontario, and Whistler Mountain, British Columbia, only daytime NPF was observed (Pierce et al., 2012; Pierce et al., 2014).

The timing of NPF is further illustrated in Figs. 6 and 7, which show the diurnal variation of the aerosol size spectra averaged over the entire measurement period and for an exemplary day at each site, respectively. At EL, Fig. 6a shows a distinct nucleation mode appearing around 23 LT, representing the nighttime events, which intensifies during the night to reach the highest $N_{nuc}$ concentrations around 06 LT. Growth into the Aitken mode continues throughout the day, and around mid-day the nucleation mode is absent. Another set of NPF events occurs between 16 and 20 LT, also followed by growth into the Aitken mode. At NR (Fig. 6b) we see overall lower total particle concentrations, much lower $N_{nuc}$ concentrations, especially at nighttime, and a relatively more prominent Aitken mode at around 70 nm. The diurnal timing





of NPF events is less distinct at NR than at EL, but there is a suggestion of elevated $N_{nuc}$ between 00 and 07 LT, and
       around 14 LT.

a)

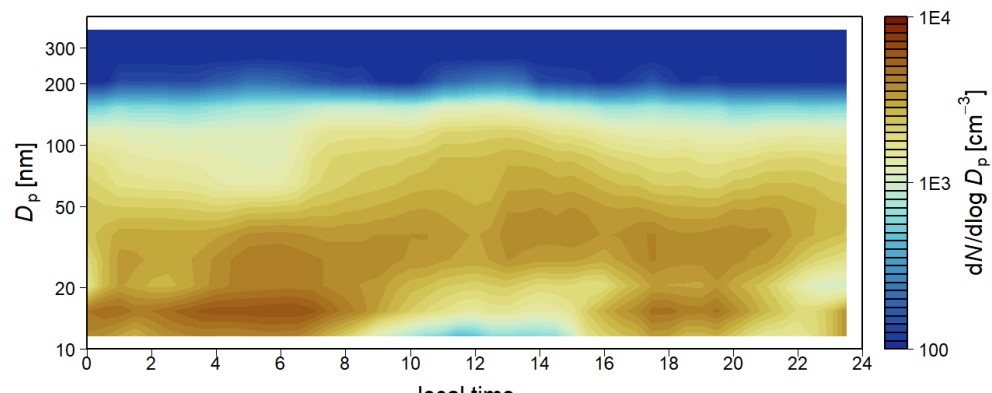

b)

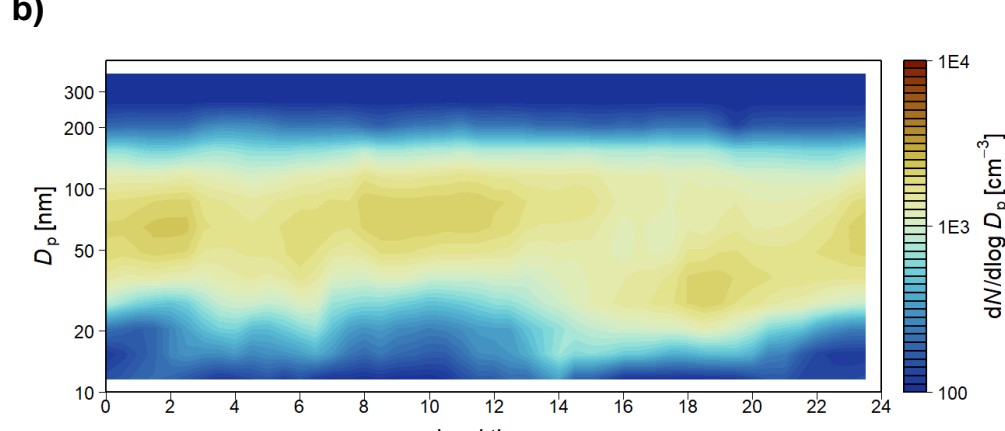

**Fig. 6: Diurnal plot of the mean number size distribution of aerosol particles at a) Eagle Lake and b) Nazko River.**

The details of the aerosol evolution are illustrated for exemplary days in Fig. 7. At EL (Fig. 7a), 11 Jun was a day with
very light winds, which resulted in a nearly stationary airmass. After midnight, particle concentrations were quite low,
with a faint nucleation mode below 20 nm, an Aitken mode around 50 nm, and a weak accumulation mode around 100

nm. A distinct new nucleation mode appears at 02 LT, which grows throughout the day to ca. 50 nm. Additional minor
       NPF events are seen in the afternoon and around 23 LT. In contrast, at NR, Fig. 7b shows the near-complete absence of
       nucleation mode particles at night, followed by a daytime NPF event with fairly rapid growth into the Aitken range.





a)

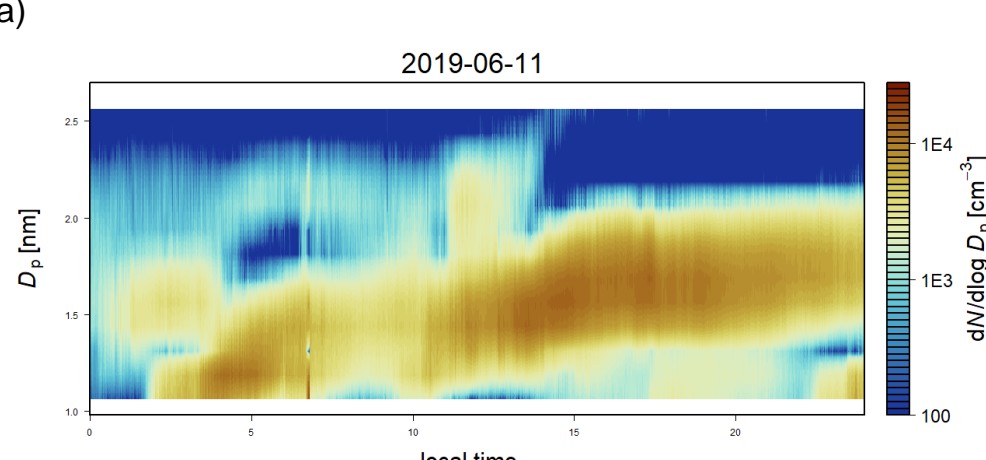


b)

Fig. 7: Time series plots of the aerosol number size distribution of aerosol particles on exemplary days at a) Eagle
Lake and b) Nazko River.

**3.4.3 Airmass origin**

As already discussed above in sections 3.2 and 3.3, most of the airmasses sampled at our sites had crossed the Pacific
coast 1 – 2 days before arriving at our sites, and contained little or no detectable anthropogenic pollution. Thus, similar
to other temperate/boreal sites, e.g., Hyytiälä and Pallas, our NPF events occurred in clean airmasses, mostly originating
from the west and northwest, which had low levels of pollution aerosols that would suppress nucleation by acting as

condensation sink (CS) (Sogacheva et al., 2005; Dada et al., 2017).

There was, however, no evidence that a marine influence on the airmass enabled or facilitated NPF at the EL site. The
airmasses arriving on 8-13 Jun had not had contact with the sea surface, as they had either remained for the last 48 hours
in the continental boundary layer (8 Jun) or had descended from the free troposphere. In these airmasses, NPF was just
as active as in the airmasses arriving on 4-7 and 14-15 Jun, which had come from the surface level over the Pacific and

had traveled in the boundary layer for their entire 48-h history.





On the other hand, even quite small levels of pollution were enough to suppress NPF. The airmasses that had arrived on 21-22 Jun from a region polluted with wildfire smoke had $M_{420}$ concentrations of only $1.5 - 3.0$ µg m$^{-3}$ and $N_{420}$ concentrations of ca. $1000 - 1500$ cm$^{-3}$ (Fig. 4b), yet even this small amount of pollution was able to completely suppress NPF.

### 3.4.4 Condensation sink

The surfaces of preexisting aerosol particles act as a sink for condensable species, such as gaseous $H_2SO_4$ or extremely-low-volatility organic compounds (ELVOCs), thereby potentially suppressing their concentrations below those required for nucleation and NPF. We estimated the CS at our sites from size distribution data from the Nanoscan and the dust and seasalt concentrations from the MERRA-2 reanalysis, using the equations in Kulmala et al. (2012). The size distribution of dust and seasalt was represented by a single mode at 3.5 µm with a geometric standard deviation of 2.0 (Albani et al.,

2014). The CS was completely dominated by the submicron aerosol, with dust and seasalt typically contributing less than 1%. For comparability with other sites, we calculated CS using the diffusion coefficient of $H_2SO_4$, although, as will be discussed below, it is more likely that the actual condensing species were organic compounds.

The CS at the start of NPF events ranged from 0.0006 to 0.0048 s$^{-1}$ (Table 1), with average values of 0.0025±0.0011 s$^{-1}$ at EL and 0.0012±0.0010 s$^{-1}$ at NR. These CS values are similar to those for the NPF event days at Hyytiälä, which

typically fell into the range of 0.002 to 0.004 s$^{-1}$ during June (Dada et al., 2017). At Pallas, event days in spring happened typically at CS in a similar range, between 0.0004 and 0.003 s$^{-1}$ (Asmi et al., 2011). With only one exception (event 18), all daytime NPF events occurred when CS was below the temperature-dependent threshold value obtained at Hyytiälä by Dada et al. (2017).

### 3.4.5 Growth rates

We calculated the growth rates (GR) of the newly formed particles based on the method of Kulmala et al. (2012) by fitting modes to the observed size distributions and deriving the time rate of change of the modal diameter during the growth phase of the NPF event. The overall range of GR during our campaign was $0.18 - 6.0$ nm h$^{-1}$, with averages of 2.1±1.2 and 3.4±1.6 nm h$^{-1}$ at EL and NR, respectively. The faster growth at NR may be related to the lower number concentrations at this site. These GR values are in good agreement with median GRs for the spring/summer seasons at the temper-

ate/boreal sites in Eurasia (Varriö, Pallas, Abisko, Tiksi, Hyytiälä, and Aspvreten), which range between 1.6 and 4.6 nm h$^{-1}$ (Nieminen et al., 2018), at Egbert, Ontario, where the mean GR was 3.1 nm h$^{-1}$ (Pierce et al., 2014), and at Whistler Mountain, BC, where GRs were 2-5 nm h$^{-1}$ (Pierce et al., 2012).

### 3.4.6 Formation rates

Given the lower cutoff of our instrument at 10 nm, we were not able to derive an estimate for the actual nucleation rate,

$J^*$. Instead, we calculated the formation rate of particles >10 nm, $J_{10}$, from the rate of increase of particle number concentrations in the size range $10 - 24$ nm during the early part of the NPF events, following the method of Kulmala et al. (2012). We applied the correction for the coagulation sink, using the parameterization given in eq. (7) of that paper to derive the coagulation sink from the condensation sink. At EL, the average value of $J_{10}$ was 0.32±0.16 cm$^{-3}$ s$^{-1}$, while at NR the formation rate was much lower, with an average of 0.12±0.11 cm$^{-3}$ s$^{-1}$. Overall, the range of formation rates

observed at our sites is in good agreement with the median values of 0.1 to 0.52 cm$^{-3}$ s$^{-1}$ for the spring/summer seasons at the Scandinavian and Siberian temperate/boreal sites listed by Nieminen et al. (2018). At Egbert, the mean $J_{10}$ was significantly higher, likely due to the substantial anthropogenic $SO_2$, and thus elevated $H_2SO_4$ vapor, concentrations at that site (Pierce et al., 2014).



### 3.5 New particle formation mechanism

In previous studies, nucleation processes based on $H_2SO_4$, iodic acid ($HIO_3$), or ELVOCs have been described as mechanisms leading to NPF in terrestrial and coastal environments. Because of the limited information obtained in our campaign, especially the lack of gas phase measurements of precursor and nucleating species, we do not have conclusive evidence regarding the mechanism of NPF at our sites. In the following paragraphs, we will examine the potential mechanisms and discuss which one of them is the most consistent with our observations.

Iodic acid has been identified as a nucleating species at coastal sites (Sipilä et al., 2016) and in laboratory studies (He et al., 2021), and may play an important role in pristine environments, where $H_2SO_4$ is at very low concentrations. It is, however, unlikely to play a role at our sites, since $HIO_3$ is formed rapidly near its precursor sources and would lead to nucleation close to the coast rather than far inland. Particles nucleated at the coast would have grown over the 1-2 days traveling to our site and would show up in the Aitken mode rather than in the nucleation mode.

Nucleation based on $H_2SO_4$ together with stabilizing species, such as ammonia, water, amines, or organics, is the most common mechanism identified at sites around the world. This mechanism, however, is active at remote and rural sites only during daytime, when sufficient $H_2SO_4$ vapor concentrations can be photochemically produced. At Hyytiälä, for example, NPF with growth into the nucleation mode occurs only during daytime (Dada et al., 2017). While nighttime nucleation events are quite common there, the new particles never grow beyond a few nanometers (Junninen et al., 2008).

In contrast, at our sites we found that there is no clear preference for daytime NPF, and that nighttime events are about as frequent as daytime ones. Frequent nighttime NPF would not be expected if marine sulfur emissions, acting as precursors of $H_2SO_4$, played an important role at our sites, as proposed by Lawler et al. (2018), who observed that NPF events at Hyytiälä occurred preferentially in airmasses that had recently been over the ocean. While most of our airmasses had also had contact with the ocean within the last few days, NPF was just as abundant at night as during daytime, and NPF 380 occurred also in airmasses that had no recent contact with the ocean.

Our observations thus argue strongly for pure organic nucleation based on ELVOC species formed from monoterpenes by autoxidation and reaction with ozone. The important role of non-photochemical processes is further supported by the lack of a clear preference of NPF events on clear-sky days, in contrast to what has been observed at Hyytiälä (Dada et al., 2017). Nighttime nucleation has also been observed at a limited number of other sites, particularly at clean sites with high 385 emissions of monoterpene precursors. For example, at Tumbarumba, in an Australian eucalypt forest, nighttime NPF events were 2.5 times as frequent as daytime events during summer/autumn (Lee et al., 2008; Suni et al., 2008). On the other hand, at Abisko, Sweden, Svenningsson et al. (2008) observed only occasional nighttime events, and at Värriö, Finland, nighttime NPF only accounted for a small fraction (16 of 147) of events (Vehkamäki et al., 2004). Interestingly, even though NPF is quite uncommon at the Siberian ZOTTO site, there is also a significant fraction of nighttime NPF 390 among these rare events (Uusitalo et al., 2021).

The ELVOC species supporting nucleation at our sites are most likely HOMs produced from monoterpenes by ozonolysis and/or autoxidation of peroxy radicals following initial attack by OH (Ehn et al., 2010; Ortega et al., 2012; Bianchi et al., 2019). The high density of conifers around our sites, especially at EL, is a prolific source of monoterpenes, to the point that their characteristic odor was perceptible on warm days, especially on 11 and 12 Jun when warm temperatures coin-395 cided with low wind speeds. These days also produced very strong and sustained NPF events (Fig. 5a).

While the relatively short observation period does not allow a detailed analysis of the relationship between NPF and ambient temperatures, our data do suggest greater production of $N_{nuc}$ at warmer temperatures. The mean $N_{nuc}$ in the cooler period from 4 to 10 Jun was $1050\pm770$ cm$^{-3}$ (5-min averaged data, N=1759), whereas during the warmer period from 11 to 15 Jun it was $1180\pm880$ cm$^{-3}$ (N=1283). While this difference is modest, it is significant at $p<0.0001$. The increase in 400 $N_{nuc}$ with higher temperatures could be caused by increased monoterpene emissions, which favor NPF and particle growth,



as has been shown previously by Kulmala et al. (2004). Temperature effects on the rate of HOM formation and nucleation are not likely to be important over the limited range of ambient temperatures during our study. Low temperatures decrease the rate of HOM formation (Frege et al., 2018), but increase nucleation probability due to lower volatility of the oxidation products, resulting in only a modest net change of the nucleation rates from organic precursors (Simon et al., 2020). While

more rapid HOM production during the warmer daytime could lead a build-up of HOMs, followed by NPF at the colder nighttime temperatures, this effect is likely to be very small for the diurnal temperature range (about 10 ℃) at our sites (Simon et al., 2020).

Other than the diurnal change of relative humidity, there were no systematic RH variations that would allow examination of the effect of RH on NPF events. Based on previous studies, no strong effects would be expected, anyway. Low RH has

been shown to favor NPF (Hamed et al., 2011; Dada et al., 2017). However, the mechanism proposed by Hamed et al. (2011) is based on reduced $H_2SO_4$ production due to lower OH at very high humidities (>80 %), which were almost never present during our study. Reduced $H_2SO_4$ production would also not affect our proposed NPF mechanism, which is based on pure organic nucleation. The increase of the CS due to hygroscopic growth is also not likely to be significant, since the particles at our BC sites are presumed to be mostly organic and thus are not expected to show strong hygroscopic

growth over the range of RH prevailing at our sites. Laboratory studies by Bonn et al. (2002) suggested that water vapor suppresses the formation of ELVOCs from monoterpenes. Note that this effect is a function of absolute, not relative humidity. Contrary to what would be expected from findings of Bonn et al. (2002), we actually observed higher $N_{nuc}$ during the (warmer) period with higher water vapor mixing ratio (11 – 15 Jun, 6.7±0.9 g kg⁻¹) than during the (cooler) period with lower water vapor (4 – 10 Jun, 4.4±0.8 g kg⁻¹).

Ozone plays a critical role as a key oxidant leading to formation of ELVOCS from monoterpenes (Ehn et al., 2014). The minimum $O_3$ levels required to initiate NPF with monoterpenes in chamber studies were 10-19 ppb (Ortega et al., 2012). While we have no on-site ozone measurements, the ozone data from nearby sites indicate that there were sufficient $O_3$ concentrations to fulfill this requirement. The $O_3$ data from the Williams Lake monitoring site (the closest site to EL) ranged between an average morning low of 13 ppb and a late afternoon high of 34 ppb (https://envistaweb.env.gov.bc.ca/,

last accessed 20 Jul 2021) for our study period (Supplemental Fig. S3). Hourly data were not available from Quesnel, the closest monitoring site to NR, but data for the daily maximum concentrations showed values similar to Williams Lake. Overall, then, our results are most consistent with a mechanism where HOMs/ELVOCs are formed by ozonolysis and/or OH-initiated autoxidation of monoterpenes followed by pure organic nucleation. The high incidence of nighttime NPF is consistent with the findings of Sulo et al. (2021), who showed that HOM dimers have their maxima during the night,

whereas the HOM monomers and $H_2SO_4$ exhibit daytime maxima. The highest concentrations of both monoterpenes and $O_3$ can be expected in the late afternoon, which may explain why we frequently observed the onset of events with particles >10 nm around midnight, which, given growth rates of 2-3 nm h⁻¹, would imply that the actual nucleation event began around sunset. Analogously, the nighttime nucleation events at Hyytiälä, typically occur around sunset and are driven by HOMs from monoterpene oxidation (Rose et al., 2018), but in contrast to our sites, at Hyytiälä the particles from nighttime

NPF never grow beyond a few nanometers (Junninen et al., 2008). Our observed growth rates, averaging 2.1 and 3.2 nm h⁻¹ for the two sites, are in good agreement with the median GRs attributable to monoterpene oxidation products (1.0 – 3.5 nm h⁻¹) measured at Pallas by Asmi et al. (2011).

The pronounced difference in NPF frequency between our two sites may be related to the difference in vegetation in the two areas. Whereas the landscape around EL is completely dominated by conifers, a large fraction of the area around NR

has been deforested in recent years either by logging or wildfires. These cleared areas are covered by a mixture of herbaceous vegetation, small conifers, and abundant aspens. Consequently, one would expect lower concentrations of monoterpenes coupled with high levels of isoprene emitted by the aspens. Suppression of NPF in an isoprene-dominated forest



environment has been observed by Kanawade et al. (2011) and investigated in the laboratory by Kiendler-Scharr et al. (2009), who attributed it to OH scavenging by isoprene. This mechanism is not likely to be important at our sites, since

it would only affect daytime nucleation, whereas at NR, nighttime nucleation is also less frequent than at EL. More likely is the alternative mechanism proposed by Heinritzi et al. (2020), wherein isoprene reduces the yield of dimer HOMs with 19 or 20 C atoms ($C_{20}$), while increasing the yield of the more volatile dimers with 14 or 15 C atoms ($C_{15}$), thereby reducing the rate of new particle formation.

## 4 Summary and conclusions

We observed a high frequency of NPF events during four weeks of measurements in June 2019 at two pristine sites in the temperate/boreal transition zone of British Columbia, Canada. At the Eagle Lake site, every day was an event day, and many days had multiple NPF events. At Nazko River, the NPF frequency was lower, yet still, 50% of days were event days. In contrast to most sites studied previously, NPF occurred as frequently during nighttime as in daytime, with 14 out of a total of 27 events taking place at night.

Airmass trajectory analysis showed that most of the sampled airmasses had arrived from the Pacific Ocean and traveled over land for 1 – 2 days before arriving at our sites. The terrestrial fetch area has an extremely low population density and no industrial activity, resulting in essentially pristine atmospheric conditions. While a marine contribution to NPF cannot be excluded at our sites due to the limited instrumentation available for our campaign, it is likely not of significance, given that there was no preference for daytime nucleation (as would be expected for $HIO_3$ or $H_2SO_4$ as nucleating species)

and that NPF events were seen just as frequently on days with no previous ocean contact.

The average condensation sink at the start of events was $0.0025\pm0.0011$ at EL and $0.0012\pm0.0010$ s$^{-1}$ at NR, well within the range where NPF has been observed at other temperate/boreal sites (Asmi et al., 2011; Dada et al., 2017; Kerminen et al., 2018). The particle growth rates, with averages of $2.1\pm1.3$ and $3.2\pm1.6$ nm h$^{-1}$ at EL and NR, respectively, were also within the range typically observed during spring/summer at temperate/boreal sites in Eurasia and North America

(Kerminen et al., 2018; Nieminen et al., 2018). Because of the relatively high lower cutoff diameter of our instrumentation (10 nm), we could not obtain the actual nucleation rate, $J^*$. The formation rates for particles at 10 nm, $J_{10}$, averaged $0.32\pm0.16$ and $0.12\pm0.11$ cm$^{-3}$ s$^{-1}$ at EL and NR, respectively, also comparable to the Scandinavian and Siberian temperate/boreal sites listed by Nieminen et al. (2018).

While the limited data available from our campaign does not allow us to draw firm conclusions about the nucleating

species responsible for NPF at our sites, several lines of evidence point to pure organic nucleation as the dominant mechanism. The strongest argument comes from the fact that nighttime NPF was as frequent as daytime NPF, analogous to other sites where organic nucleation has been shown to dominate, and ruling out photochemical production of $H_2SO_4$ as source of nucleating species at least during the night. The lack of anthropogenic sources of $SO_2$ and the independence of NPF from marine influence on the sampled airmasses also argues against nucleation driven by $H_2SO_4$. Finally, the pres-

ence of abundant monoterpene sources in the fetch (to the point that they sometimes could be detected by their odor) and the potential dependence of the NPF frequency on the presence of isoprene-emitting vegetation also support organic nucleation as the dominant mechanism.

Our results are consistent with the model-derived importance of pure organic NPF in remote regions (Gordon et al., 2017; Zhu and Penner, 2019), however, they raise important questions about the extent to which pristine NPF conditions still

exist in the present-day atmosphere. At the vast majority of remote sites studied so far, NPF is exclusively a daytime phenomenon, suggesting dominance of $H_2SO_4$ as the controlling species. Would these sites have been dominated by pure organic nucleation in pre-industrial times, with frequent nighttime NPF? And, why is NPF so infrequent at the remote





subboreal ZOTTO site in central Siberia, which is surrounded by vast coniferous forest (Wiedensohler et al., 2019; Uusitalo et al., 2021)? Possibly, the large distance to anthropogenic sources of $SO_2$ has allowed it to be fully converted

to sulfate aerosol, removing the source of $H_2SO_4$ while providing a condensation sink that prevents organic nucleation. Is there a sequence of regimes, where at truly pristine conditions, pure organic nucleation dominates, followed by $H_2SO_4$-driven nucleation with organic-dominated growth in the presence of small amounts of anthropogenic $SO_2$, again followed by a "nucleation valley of death" where the CS from pollution aerosols suppresses nucleation, and finally the highly polluted regime where there is so much $SO_2$ that $H_2SO_4$-driven nucleation can overcome the suppression by the elevated

CS?

Our study shows that there is a need for in-depth investigations at pristine continental sites to address these questions. A full set of instrumentation to identify nucleating species and precursors is required to elucidate nucleation and growth mechanisms. Also, our measurements were limited to a relatively short period in late spring to early summer, when previous studies at other sites have shown that both NPF events and HOM dimer precursor concentrations have their

seasonal maxima (Nieminen et al., 2018; Sulo et al., 2021). Future studies should include the development of a site for continuous long-term observations to investigate the seasonal and interannual variability of NPF over the remote North American forest regions.

**Author contributions**

MOA designed the experiments and MOA and TWA carried them out. MOA and FD performed the data analysis. MOA

prepared the manuscript with contributions from all co-authors.

**Competing interests.**

The authors declare that they have no known competing interests that could have influenced the work reported in this paper.

**Acknowledgments**

We thank the hosts at our sites, Clay and Marilyn Hett at Eagle Lake, and Curtis and Theresa Sharp at Nazko River, for their hospitality and support. Some analyses and visualizations used in this paper were produced with the Giovanni online data system, developed and maintained by the NASA GES DISC. We thank Zengxin Pan for help with downloading the MERRA-2 data. This research was funded by the Distinguished Scientist Fellowship Program of King Saud University and the Max Planck Society.

**Data Availability**

The Nanoscan SMPS data is archived on the Edmond data base at https://dx.doi.org/10.17617/3.7w. The Modern-Era Retrospective analysis for Research and Applications, version 2 (MERRA-2) data used is available at https://disc.gsfc.nasa.gov/datasets?project=MERRA-2.



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
