# Peer review of "Frequent new particle formation at remote sites in the temperate/boreal forest of North America"

_Atmospheric Chemistry and Physics, 2021_

## Author Comment (AC1)

Reviewer comments are in italics, our responses in plain font.

*This manuscript investigates atmospheric new particle formation (NPF), a topic that has been of wide interest during the recent years. Although a large number of papers on NPF in forested environments has been published, and although many of these previous investigations are based on much larger measurement data sets as employed here, I think that this paper manages to bring up some new idea on the topic by focusing on differences between pristine and anthropogenic-influenced forested environments. The paper appears to be scientifically sound in terms of both applied methodology and performed analysis. Although somewhat speculative, the paper is in general very well written. I recommend accepting this paper for publication after the authors have addressed the following, mostly minor issues.*

We thank the reviewer for these positive comments. We would like to emphasize that while there are numerous studies on NPF in the Eurasian region, there are very few in North America, and none before ours had been performed in an environment approximating pristine conditions.

*Scientifically, the weakest feature of this paper is the shortage of data (less than one month of measurements), on which to base all the made conclusions. Previous long-term studies have pointed out a considerable variability in the characteristics of NPF from day to day, seasonally, and even between different years at a single measurement site. Therefore, one needs to be very careful how much to conclude from this data set. I understand that the number of data cannot be increased at this stage, and that the purpose of this paper is to raise up new scientific issues rather than to make firm conclusions. I also appreciate that the authors have tried to take the shortage of data into consideration when discussing the results. However, I still think that the authors should bring up this issue more explicitly throughout the paper when discussing the results. This would benefit specifically those readers that not very familiar with this research topic.*

As the reviewer correctly states, the purpose of our study was not to be the final word on NPF in North America. Rather, we wanted to conduct a pilot study in what we believe to be one of the most pristine subboreal forest environments in the Northern Hemisphere, in order to stimulate further, more comprehensive and long-term studies on this issue. We had already alluded to the limitations of our data set in the original manuscript, e.g., l. 169-170 (old): "Given the relatively short duration of our study compared to the long-term studies at some other sites, we can only make limited comparisons regarding NPF frequency at our BC sites."

In addition, we now state at the beginning of the conclusions: "We conducted a pilot study on NPF at two pristine sites in the temperate/boreal transition zone of British Columbia, Canada, extending over four weeks of measurements in June 2019. At both sites, we observed a high frequency of NPF events."

In the last paragraph of the Conclusions, we again emphasize the limited duration of our campaign, and highlight the need for longer-term studies.

Based on a suggestion by Reviewer 2, we also made a comparison between the meteorological conditions during our campaign with the long-term averages and found no substantial bias. We

are now also including land cover maps (Figs. 1a and 1b) that show that our sites are surrounded by vegetation typically of the region.

*The end of section 1 lacks clear scientific objectives of this study. They should be added, especially when considering that the other parts of the introduction are very well written and informative.*

We added the following text at the end of section 1:

"The objectives of our study were (1) to determine whether NPF in a pristine subboreal forest environment was frequent, like at the Scandinavian sites, or almost non-existent, like at the Central Siberian sites, (2) elucidate the role of anthropogenic $SO_2$ emissions in NPF by making measurements in a region were such emissions were likely to be negligible, (3) examine the hypothesis that nighttime NPF would be frequent in the absence of significant sources of $H_2SO_4$ vapor, and (4) examine whether the results from such a limited pilot study would warrant future, more comprehensive and extended studies."

*Minor issues*

*The authors use 2-day air mass trajectories, yet make conclusions to 3 days of air mass transport back in time (line 205). Is this based on extrapolation of available air mass trajectories or what? Some justification here is needed.*

We based our analysis of airmass history on both 48-hour and 10-day backtrajectories. This was maybe not clearly enough stated in the text. We now added the following sentence at the beginning of section 3.2:

"We investigated the history of the sampled airmasses using 48-hour and 10-day backtrajectories initialized 100 m above surface level at 12 local time (LT = 19 UTC)."

*Lines 218-222: Saying that observed mass concentrations are extremely high and number concentration extremely low is a vague statement. Extreme compared to what? And are they quantities really extremely high/low?*

We replaced "extremely" by "very" and added comparisons and references here:

"The mass concentrations, $M_{420}$, are very low (Fig. 4a), with an average of 0.73 (range 0.11 – 2.3) $\mu g\ m^{-3}$, ten times lower than the average PM2.5 value of 7.3 $\mu g\ m^{-3}$ for North America (Mortier et al., 2020). The BC, dust, and seasalt concentrations from MERRA-2 are also very low, at $0.041\pm0.011$, $2.9\pm1.5$, and $0.41\pm0.37\ \mu g\ m^{-3}$, respectively, for the measurement period (Supplemental Fig. S2a). In contrast, the particle number concentrations, $N_{420}$, are very high compared to other pristine sites (Andreae, 2009), with an average of 3150 (850 – 10300) $cm^{-3}$, of which a large fraction are in the nucleation mode below 24 nm (average 1100, range 55 – 5250 $cm^{-3}$) (Fig. 4a)."

We did not want to go into a lengthy set of comparisons here, but for example at our Siberian site under clean conditions, mass concentrations are typically two to three times higher. Similar low concentrations to those we measured in BC are seen at the remote rainforest site ATTO in the Amazon only during the very cleanest time in the rainy season.

*Figure 1 is not very helpful in its current form. The names of the locations/places in the maps are barely visible and should be written using larger font sizes. The maps should also include length scales, especially as the scale seems to vary from map to map.*

We recreated these maps with larger fonts and length scales. In addition, we replaced Figs 1b and 1c with land cover maps and included pie charts that represent the land cover types in the fetch areas.

---

## Author Comment (AC2)

Reviewer comments are in italics, our responses in plain font.

This manuscript summarizes new measurements of aerosol size distributions made in summer in pristine British Columbia at two sites 150km apart. The measurements were short in duration and not simultaneous. One site exhibited very frequent new particle formation, often at night, while the other did not. The authors have made the most important datasets on the aerosol size distribution publicly available. They make interesting and useful comparisons of the sites to other places where similar new particle formation is observed or may be expected. The paper motivates more measurements in this very interesting pristine region where conditions are likely not far from pre-industrial. The topic is important, the paper is well written and deserving of prompt publication. I have only a very few minor comments.

Abstract: would it be worth discussing briefly the very interesting diurnal cycle of new particle formation?

We added the following sentences to the Abstract: "In contrast to observations in other temperate/boreal environments, we found that NPF at our sites occurred at nighttime just as frequently as during daytime. Together with the lack of identifiable sources of  $H_2SO_4$  precursor species in the fetch region of our sites, this suggests that nucleation of extremely-low-volatility organics was the predominant NPF mechanism."

Methods: was the SW radiation shown in Figure 1 measured or from reanalysis?

The SW radiation in Fig. 2a was from reanalysis. This information was added to the Figure caption.

The description of how the nucleation and growth rates presented in Table 1 were calculated might belong better in a short subsection in the Methods, since the Table is given much before the description in section 3.4.6?

We moved the procedures to identify NPF events and to calculate condensation sink, growth rates and formation rates to the Methods section.

I agree with Reviewer 1 that the short duration of the measurements do lead to uncertainties in the interpretation of the data. The authors do point this out already, but more discussion (based on reanalysis, for example) of how representative the weather conditions during the measurement period were of the usual conditions during the summer season would be beneficial. Such a discussion could be used to give more confidence in the results if conditions were representative, or to highlight the uncertainty if they were not.

We thank the reviewer for this suggestion. We compared the meteorological data for June 2019 in our study region with the June averages for 2000 to 2021 using MERRA-2 reanalysis and MODIS data and found no significant bias. We included a discussion on this in Section 3.1, and provide the results in Supplementary Table 1 and Supplementary Figure 1. We are now also including land cover maps (Figs. 1a and 1b) that show that our sites are surrounded by vegetation typically of the region. We added a sentence in the Summary and Conclusions section: "While

the limited duration of our study does place limits on the generalization of our results, the fact that the meteorological conditions during our study were typical of long-term average conditions at this time of year and that the land cover surrounding our sites was typical of the region does support that our results were not subject to bias from unusual weather or vegetation cover."

---

## Author Response (AR2)

*Me made the following changes in response to the editor's comments:*

1) Figure 1a is now more clear. However, please make the text Vancouver more visible as it is a well-known location that helps the reader to place the map into a global perspective.

*Done*

2) Figure 1b and 1c: The red font is not readable against the dark background. The contours are not well visible either. Please consider zooming out to show more of the footprint area.

*The figure has been modified to be more clear. We prefer not to zoom out, since we want to show clearly the land cover in the region of 100 km or so surrounding the sites, which has the greatest influence. The trajectories in Fig. 3 provide larger scale information.*

3) Figure 2a and 2b: temperature and RH timeseries cannot be read by a colorblind person. Same for Figure 4a and 4b lower panels for size fractionated number concentrations. Logscale in y-axis would bring out better the variability in low number concentrations.

*Done. We prefer the linear scale, since we want to highlight the high concentrations, which relate to NPF, the focus of our paper, rather than exaggerating the variability in low concentrations, which don't relate to the NPF events.*

4) Figure 7: Evening time there seems to be a staggered upper limit in sizes > 200 nm. Please double check.

*This is a result of the relatively wide channels of the Nanoscan compared to the TSI SMPS.*